# Multifunctional graphene heterogeneous nanochannel with voltage-tunable ion selectivity

Shihao Su [1,2,6], Yifan Zhang[1,2,6], Shengyuan Peng[1,2], Linxin Guo[1,2], Yong Liu[1,2], Engang Fu [1], Huijun Yao[3,4], Jinlong Du[5], Guanghua Du [3,4] & Jianming Xue [1,2]

Ion-selective nanoporous two-dimensional (2D) materials have shown extraordinary potential in energy conversion, ion separation, and nanofluidic devices; however, different applications require diverse nanochannel devices with different ion selectivity, which is limited by sample preparation and experimental techniques. Herein, we develop a heterogeneous graphene-based polyethylene terephthalate nanochannel (GPETNC) with controllable ion sieving to overcome those difficulties. Simply by adjusting the applied voltage, ion selectivity among $K^+$, $Na^+$, $Li^+$, $Ca^{2+}$, and $Mg^{2+}$ of the GPETNC can be immediately tuned. At negative voltages, the GPETNC serves as a mono/divalent ion selective device by impeding most divalent cations to transport through; at positive voltages, it mimics a biological $K^+$ nanochannel, which conducts $K^+$ much more rapidly than the other ions with $K^+$/ions selectivity up to about 4.6. Besides, the GPETNC also exhibits the promise as a cation-responsive nanofluidic diode with the ability to rectify ion currents. Theoretical calculations indicate that the voltage-dependent ion enrichment/depletion inside the GPETNC affects the effective surface charge density of the utilized graphene subnanopores and thus leads to the electrically controllable ion sieving. This work provides ways to develop heterogeneous nanochannels with tunable ion selectivity toward broad applications.

Ion selective artificial nanochannels have recently attracted general interest in water desalination[1,2], ion separation[3,4], and nanofluidic devices[5,6]. The boom of 2D materials such as graphene and molybdenum disulfide ($MoS_2$) makes nanoporous atomically thin membranes (NATMs) highly promising as next-generation ion-selective membranes, owing to their ultra-thin material thickness, high mechanical strength, and great chemical robustness[7,8]. These nanochannel devices are supposed to have different ion selectivity in different situations. In energy harvesting scenario, it is crucial for ion-exchange membranes with high cation/anion selectivity to achieve the high conversion efficiency and output power density[9,10]. Rapid and selective conduction of single ions (e.g., $K^+$) is concerned in a biomimetic ion transport channel[11,12]. The industry of lithium-ion batteries needs efficient extraction of $Li^+$ from brine or seawater with the exclusion of other ions[13,14]. Under each circumstance, a corresponding device with specific ion selective nanochannels is required to be fabricated, which could be time- and resource-consuming and limited by experimental techniques.

[1]State Key Laboratory of Nuclear Physics and Technology, School of Physics, Peking University, Beijing 100871, PR China. [2]CAPT, HEDPS and IFSA, College of Engineering, Peking University, Beijing 100871, PR China. [3]Institute of Modern Physics, Chinese Academy of Sciences, Lanzhou 730000, PR China. [4]University of Chinese Academy of Sciences, Beijing 100049, PR China. [5]Electron Microscopy Laboratory, School of Physics, Peking University, Beijing 100871, PR China. [6]These authors contributed equally: Shihao Su, Yifan Zhang. ✉e-mail: gh_du@impcas.ac.cn; jmxue@pku.edu.cn

NATMs with tunable ion selectivity are desirable to overcome those difficulties and have gained widespread attention in the last few years[12,15–23]. For example, Abraham et al.[15] prepared graphene oxide (GO) membranes with different interlayer spacing by adjusting the relative humidities of the sealed storage containers, and those GO sheets exhibited different relative cation permeation rates. Using a time-dependent nitrogen-doping technique, Song et al.[16] created several nitrogen-doped nanoporous graphene with different ion sieving abilities. Zhang et al.[17] reported that the GO membranes with controllable surface charges introduced by polyelectrolyte modification exhibited various ion selectivity and charge-guided ion transport. Also, Sun et al.[18] fabricated hybrid GO-based membranes with different surface charges to achieve different charge-sensitive sieving of cations. However, in those works, a single functional membrane was actually fabricated in each case, and its ability to sieve ions was still untunable, which is the same for NATMs with diverse structures (e.g., interlayer spacing, pore diameter, surface charge, functional group at pore edge) created using different methods[17–21]. A nanochannel device with pH- or electrical-tunable ion selectivity could be promising[20,22–29]. Some nanoporous membranes had pH-dependent ion selectivity possibly due to pH-sensitive electric charges[20,22,23], but their performances might degrade in the case of given solutions with fixed pH value and they could not fast tune the ion selectivity for some industry demands. Although controllable water permeation and ion flow through nanochannels could be realized by electrically gating[24–29], experimental research on ion selectivity is rare and the mechanism remains elusive (e.g., determined by nanobubbles[28] or membrane surface charges[29]).

Ion flow can also be tuned by applied voltage in a nanochannel with ion rectification, which is the directional transport of the ions through the channel at different voltages[30,31]. In a rectifying nanochannel, such as a conically shaped polyethylene terephthalate nanochannel (PETNC), which is promising as a biomimetic nanochannel due to its flexibility and ease of chemical modification, the cation/anion selectivity can be achieved[32–34], but it is difficult for a conical PETNC with entrance diameters larger than several nanometers to sieve metal ions[35–37], because such sizes are much larger than the ion hydrated diameters (a few angstroms[7,38]) in electrolyte solutions and therefore the factors leading to ion separation (e.g., steric hindrance, electrostatic repulsion, and dielectric effects such as dehydration)[3] hardly work. Nevertheless, the voltage-dependent ion transport in a rectifying nanochannel suggests that a nanochannel device with heterogeneous structure consisting of the pristine 2D nanopores with unchangeable ion selectivity along with the rectifying nanochannel might achieve voltage-tunable ion sieving. Such studies are few because NATMs are usually supported by non-selective and non-rectifying nanoporous materials such as $SiN_x$ and anodized aluminum with μm-sized cylindrical nanochannels, which are merely supports with negligible influence on the ion sieving[1–8].

Here, we design a heterogeneous GPETNC with graphene subnanopores on the base side of a conical PETNC. By taking the best of both the components, it exhibits excellent voltage-controlled ion selectivity among $K^+$, $Na^+$, $Li^+$, $Ca^{2+}$, and $Mg^{2+}$, which is capable of quickly meeting different demands for complex and variable scenarios in practice. The GPETNC is also able to rectify ion currents with opposite rectification for mono- and divalent ions. Finite element calculations indicate that the tunable ion selectivity arises from the different effective surface charges of graphene subnanopores influenced by the voltage-dependent enrichment/depletion of ions inside the GPETNC. The opposite preferential transport directions of the mono- and divalent cations arise from the differences in the diffusion currents, which are generated by the ion concentration gradients. The multifunctional GPETNC exhibits its promise as not only a mono/divalent ion selective device but also a biomimetic ion channel, as well as a cation-sensitive diode, which indicates the great potential of such intriguing heterogeneous nanochannels for diverse applications.

## Results
### Fabrication of the GPETNC
The GPETNC consisted of graphene subnanopores and a single conical PETNC, of which the base side was in contact with the graphene (Fig. 1a, b). The GPETNC was designed to take the advantage of both 2D nanopores and rectifying nanochannels, with their primary functional areas (graphene subnanopores and the tip entrance of the conical PETNC) spatially separated for better performances. Such structure was different from the previous structures of nanoporous 2D materials on ordinary supports (which hardly have influences on ion selectivity) with only a single confined functional area (i.e., 2D nanopores)[1–8]. An asymmetric track-etching technique[39,40] was used to fabricate the conical nanochannel in a 12-μm-thick PET membrane. Briefly, a pristine PET membrane was first irradiated by a single swift heavy ion, which could create a nanometer-sized damage track in the membrane, and then asymmetrically etched with one side of the membrane contacting the etchant and the other contacting the stop medium. The fabricated conical PETNC had the base entrance diameter of ~840 nm estimated from the etching time and the tip diameter of ~140 nm obtained by fitting the size of a theoretical model to experimental data (Supplementary Fig. 1). Monolayer graphene grown by a chemical vapor deposition method[41,42] with negligible defects (Supplementary Fig. 2) was transferred on the prepared PET membrane (making graphene on the base side of the PETNC) using a wet transfer procedure with polymethyl methacrylate as the mediator[9,43]. Ultimately, irradiation of energetic ions[9,43] was utilized to introduce subnanopores in the transferred graphene suspended on the PETNC. More details about the fabrication procedures of the GPETNC are presented in "Methods". Aberration-corrected scanning transmission electron microscope (STEM) was used to characterize the graphene subnanopores created under the ion irradiation, and the pores had the average diameter of ~0.5 nm and density of ~$5.47 \times 10^{11}$ cm$^{-2}$ (Supplementary Fig. 3), which

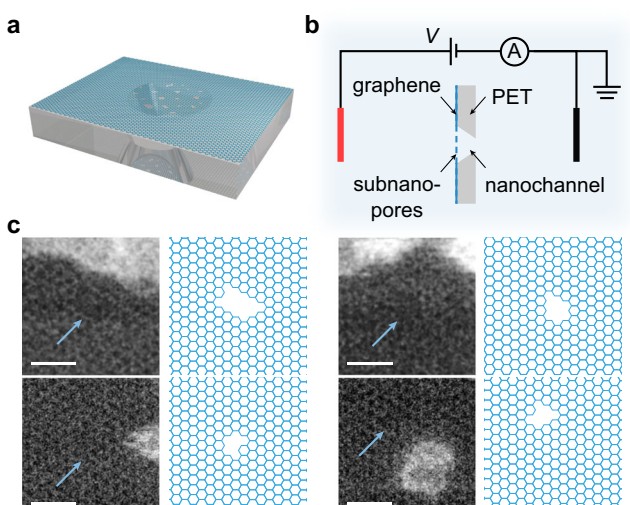

**Fig. 1 | Structure of graphene-based polyethylene terephthalate nanochannel (GPETNC) and experimental setup. a** Three-dimensional structural model of the GPETNC membrane, which consisted of nanoporous monolayer graphene and polyethylene terephthalate membrane with a conically shaped nanochannel. **b** Cross-sectional schematic illustration of the GPETNC immersed in electrolyte solutions while measuring ion currents under different applied voltages using Ag/AgCl electrodes. The graphene with subnanopores was on the base side of the polyethylene terephthalate nanochannel. **c** Cs-corrected high-angle annular dark field-scanning transmission electron microscopy (HAADF-STEM) images of representative graphene subnanopores fabricated under ion irradiation. The pores are indicated by blue arrows, and their corresponding atomic structures are illustrated in the right panels. Scale bars are 1 nm.

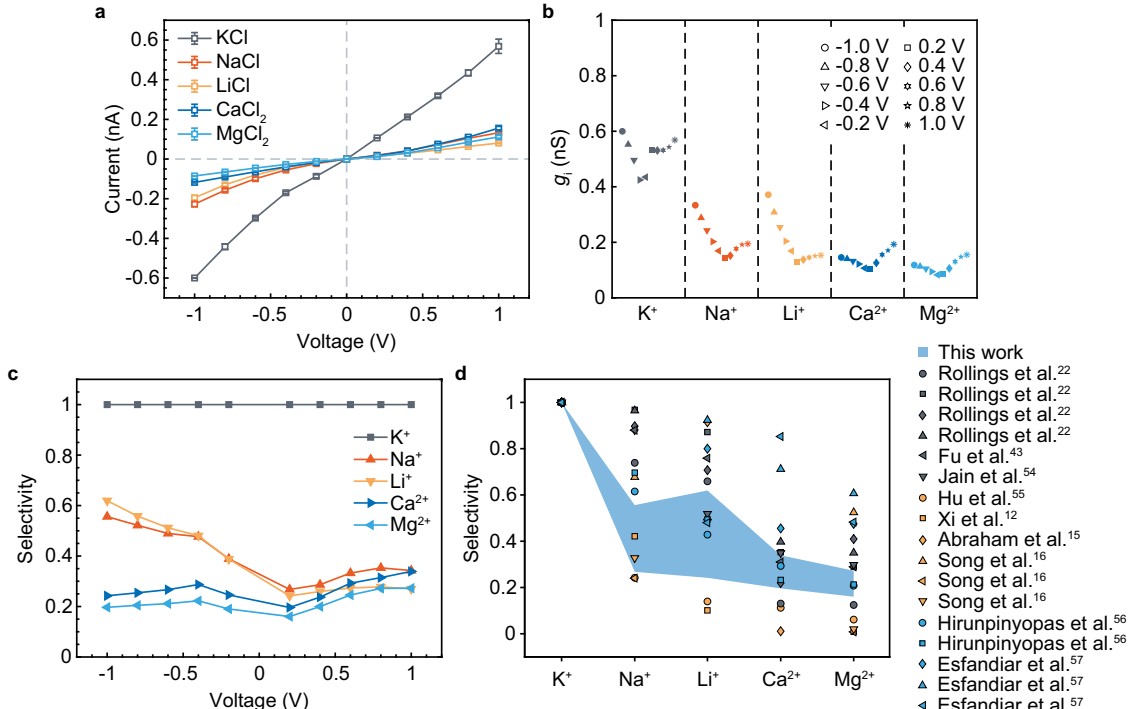

**Fig. 2 | Tunable ion selectivity of graphene-based polyethylene terephthalate nanochannel (GPETNC).** **a** *I–V* curves of the GPETNC in different chloride salt solutions with concentrations of 0.1 M. Error bars were the standard deviations from the average of three independent records. **b** Normalized conductance $g_i$ of cation i for the GPETNC at different voltages. **c** Ion selectivity of the GPETNC at different voltages. **d** Comparison of ion selectivity of the GPETNC and other nanoporous atomically thin membranes (NATMs) in experiments. The selectivity of different cations was normalized by that of K[+]. Based on the type of experimental settings, selectivities exhibited here were divided into three groups:

conductance (gray), permeation (yellow), and mobility (blue), which indicated that they were obtained by comparing the normalized ion conductance[22,43,54], the ion permeation rate[12,15,16,55], and the mobility ratio of cation against chloridion[56,57], respectively. The light blue region is the range of tunable selectivity of the GPETNC at different voltages in this work (see Supplementary Fig. 5 for the original data). The NATMs in references include graphene[16,22,43,54], graphene oxide with various modifications[12,15,16,55], molybdenum disulfide (MoS$_2$)[56], and composite structures of graphene, MoS$_2$, and hexagonal boron nitride (hBN)[57].

are consistent with the results of our recent molecular dynamics simulations[44]. Figure 1c shows four representative subnanopores with the corresponding atomic structures, and the observed pores (see all of them in Supplementary Fig. 3) had structures in agreement with the theoretical predictions made by Rajan et al.[45] when solving the isomer cataloging problem of nanopores in monolayer graphene.

**Tunable ion sieving in the GPETNC**

To measure the ion selectivity, the GPETNC was characterized by recording the current-voltage (*I-V*) curves in a custom-designed system with two cells filled with chloride salt solutions using Ag/AgCl electrodes (Fig. 1b). We first considered the cation/anion selectivity of the GPETNC. Graphene nanopores created under ion irradiation have been reported to exhibit extremely high cation/anion selectivity[9,43], which means that most anions are impeded when transporting through the pores compared to cations. The cation/anion selectivity arises from the negative charges on graphene nanopores during its transfer or fabrication in experiments, despite the debate of the nature of electric charges on nanoporous graphene surface due to hydroxide adsorption or pore edge due to functional groups in electrolyte solutions[22,43,46–49]. Hence, the GPETNC was likewise supposed to have negative charges on the graphene subnanopores and therefore be capable of preventing the transport of anions. We further identified the K[+]/Cl[−] selectivity of the GPETNC to be 46 using the Goldman–Hodgkin–Katz model[50,51] (see details in "Methods"), which does not explicitly depend on the channel size and has been successfully applied in studying the cation/anion selectivity of graphene nanopores[9,22,43]. The high selectivity ratio confirmed that the transport of Cl[−] through the GPETNC could be neglected compared to that of cations.

Inter-cation selectivity of the GPETNC was obtained by directly comparing the normalized conductance $g_i$ of cation i (compared to that of K[+]), which was the measured ion conductance in 0.1 M cation-chloride solutions (KCl, NaCl, LiCl, MgCl$_2$, and CaCl$_2$) adjusted to consider the differences in electric mobilities and charges of the cations (as shown in "Methods"). The selectivity $S_i$ defined using this method is effective to describe the ability of the GPETNC to sieve different ions (a larger $S_i$ indicates that cation i can transport more easily) when the ion current caused by Cl[−] is negligible[22,43]. Figure 2a shows the *I–V* curves of the GPETNC in different chloride salt solutions, which are nonlinear and different from the results of pristine graphene nanopores[22,43]. Therefore, the normalized conductance $g_i$ was calculated at different voltages (Fig. 2b), and the number was found to be voltage-dependent and varied with different ions, which indicates the diverse abilities to transport through the GPETNC.

The ion selectivity of the GPETNC is presented in Fig. 2c. At negative voltages, $S_{K^+} > S_{Na^+} \approx S_{Li^+} > S_{Ca^{2+}} \approx S_{Mg^{2+}}$, which is similar to the order of the ion selectivity of pristine graphene nanopores[22,43]. In this case, the GPETNC exhibited the ability to sieve the mono- and divalent ions with mono/divalent ion selectivity up to ~3.3 (the average selectivity of monovalent ions against that of divalent ones at −1 V). In addition, the ion selectivity was voltage-dependent, especially for monovalent ions (Na[+] and Li[+]), whose selectivity decreased with the increase of applied voltage.

However, at positive voltages, we surprisingly found that the GPETNC exhibited $S_{K^+} \gg S_{Na^+} \approx S_{Li^+} \approx S_{Ca^{2+}} \approx S_{Mg^{2+}}$ with the maximum K[+]/ions selectivity of about 4.6 at 0.2 V (Fig. 2c), which is different from the results at negative voltages. Under this circumstance, the GPETNC conducted K[+] up to ~4.6 times more rapidly than the other

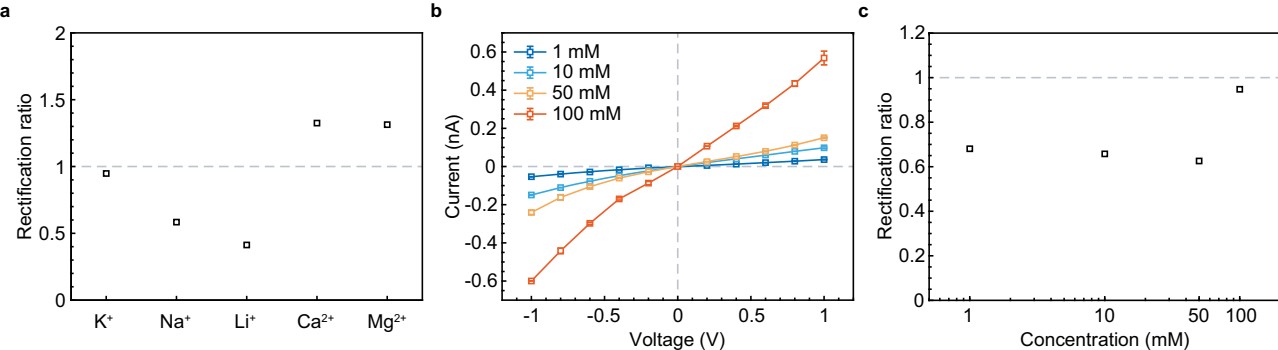

Fig. 3 | **Rectification of graphene-based polyethylene terephthalate nano-channel (GPETNC). a** Rectification ratio of the GPETNC in different cation-chloride solutions with concentrations of 0.1 M. **b** I–V curves of the GPETNC in KCl solutions with different concentrations. Error bars were the standard deviations from the average of three independent records. **c** Rectification ratio of the GPETNC in KCl solutions with different concentrations.

ions, which well resembles a biological potassium-selective ion channel, where the ion transport tends to be affected by the transmembrane voltage[52,53]. Therefore, the results suggest fresh ways to mimic biological ion channels and study the in-channel ion transport mechanism using artificial nanochannels with similar heterogeneous structures. The above findings indicated that the ion selectivity of the GPETNC could be facilely tuned by simply adjusting the applied voltage, and the GPETNC is a promising multifunctional nanochannel as a mono/divalent ion selective device at negative voltages and a biomimetic K$^+$ channel at positive voltages.

We further compared the ion selectivity of the GPETNC with that of some NATMs reported before[12,15,16,22,43,54–57] as shown in Fig. 2d. The light blue region is the range of the tunable ion selectivity of the GPETNC in this work, in contrast with other NATMs with fixed ion selectivity (patches), and the sieving of K$^+$/ Na$^+$(Li$^+$) could be accomplished by the GPETNC while many devices fail in this. The results indicate that the great promise of the GPETNC as a versatile ion selective nanochannel because its ion selectivity can be quickly adjusted by adjusting the applied voltage, with no need in additional modifications to the sample itself. Furthermore, the GPETNC suggests such a heterogeneous nanochannel, in which, for example, nanoporous 2D materials such as MoS$_2$, hBN, GO, and early transition-metal carbides and carbonitrides (MXenes) combined with the conical PETNC, might also exhibit a facilely tunable ion sieving ability.

The ion selectivity of the GPETNC was demonstrated based on the experiments in single electrolyte solutions. We noticed that ion selective nanochannels and membranes are designed and fabricated in order to function in mixed solutions, i.e., to separate a single (or more) ion from a mixture of ions. Therefore, such research is valuable for each emerging ion selective nanochannel. However, for nanochannels with selectivities based on the experiments of ion conductance[22,43,54] or mobility[56,57], which is characterized by the measured ion current, it is difficult to study their performance in mixed solutions, because all ions contribute to the total current and their portions are hard to be distinguished. Taking K$^+$ and Na$^+$ as an example, we fabricated a GPETNC with ion selectivity shown in Supplementary Fig. 6a, and the I–V curve in the mixed solution containing both K$^+$ and Na$^+$ was complicated and indicated the elusive effect of the interaction between K$^+$ and Na$^+$ on the ion transport (Supplementary Fig. 6b). Considering that the experiments in single solutions focusing on the differences in ion conductance/mobility are effective to characterize the ion selectivity of nanochannels and that the selectivity is suggested to be valid in mixed solutions[22,43,54,56,57], detailed studies of the interaction between different ions in mixed solutions are beyond the scope of this work, and future research could focus on this.

In addition to the voltage-tunable ion sieving, the GPETNC also exhibited the ability to rectify ion currents as indicated by the

nonlinear I–V curves of the GPETNC in different chloride salt solutions (Fig. 2a). The extent of the ion rectification is generally characterized by the rectification ratio (i.e., the ratio of the absolute ion currents at opposite voltages according to the I–V curves, which was defined as $|I_{+1V}|/|I_{-1V}|$ here), and its value also reflects the preferential direction of ion transport in rectifying nanochannels well[31,32,58,59]. Figure 3a shows the rectification ratios of different ions of the GPETNC. For monovalent ions (K$^+$, Na$^+$, and Li$^+$), their rectification ratios were smaller than 1, which indicated that the transport of these ions in the GPETNC had a preferential direction from the tip side to the base side. However, for divalent ions (Ca$^{2+}$ and Mg$^{2+}$), the preferential direction of transport reversed as base-to-tip, as the rectification ratios were found to be larger than 1. Although ion rectification phenomenon exists in a pristine conical PETNC in different chloride salt solutions at the same concentration[35–37], the rectification ratios are always smaller than 1 for not only monovalent ions but also divalent ones, which indicates that the preferential direction of all of them keeps from the tip to the base entrance. Besides, the preferential direction does not reverse in electrolyte solutions with different concentrations for a pristine conical PETNC[35–37], and neither did the GPETNC (see Fig. 3b, c). Hence, the opposite rectification of mono- and divalent ion transport was peculiar to the GPETNC, and thus it could serve as a cation-responsive nanofluidic diode.

## Heterogeneous structure of the GPETNC determines its tunable ion selectivity

The performance of a nanochannel can be attributed to its structure[60], and the same also applies for the GPETNC with the heterogeneous structure of graphene subnanopores on the base side of a conical PETNC (Fig. 1). The ion selectivity of the GPETNC is attributed to the existence of the graphene subnanopores, because a pristine conical PETNC has entrance diameters much larger than the ion hydrated diameters and therefore it can hardly sieve metal ions[35–37]. However, it is reported that the inter-cation selectivity of pristine graphene nanopores is independent on the applied voltage[22,43], even for a cylindrical PETNC support[43], which indicates that the voltage-tunable property of the ion selectivity of the GPETNC arises from the use of the conical PETNC (Fig. 1). The main difference between the conical and cylindrical PETNCs is that the former has voltage-dependent ion enrichment/depletion effects inside the channel while the latter does not[30–32]. Hence, the ion sieving of the GPETNC originates from the graphene subnanopores, and it is determined by certain factors (such as the diameter, structure, charge density, and functional groups of the pores)[3,4,7,8], which we speculated to be affected by the voltage-dependent enrichment/depletion of ions inside the GPETNC to some extent.

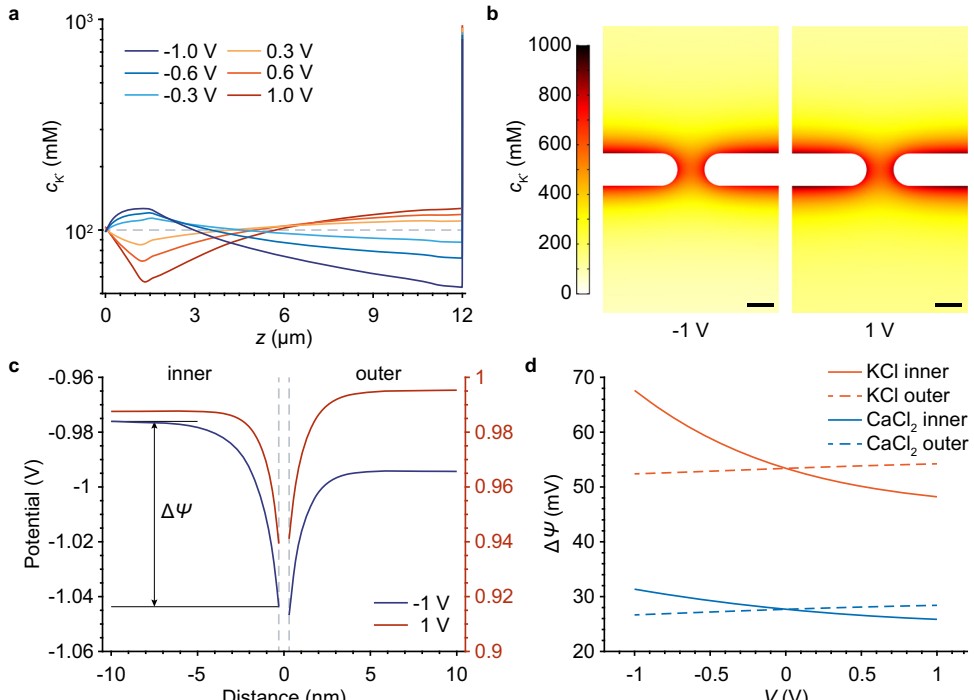

**Fig. 4 | Numerical calculations of distributions of ion concentration and electric potential in graphene-based polyethylene terephthalate nanochannel (GPETNC). a** Concentration of potassium ion $c_{K^+}$ along the center symmetry axis $z$ of the GPETNC in 0.1 M KCl solution at different voltages. $z = 0$ and 12 µm represent the tip and base side of the GPETNC, respectively. **b** Distributions of $c_{K^+}$ nearby a graphene subnanopore of the GPETNC in 0.1 M KCl solution at −1 and 1 V. Scale bars are 0.5 nm. **c** Electric potential $\psi$ at different distances away from the graphene center along a perpendicular path nearby a graphene subnanopore of the GPETNC in 0.1 M KCl solution at −1 and 1 V. The path is 0.75 nm away from the pore center (Supplementary Fig. 8e). The gray dashed lines represent the graphene surfaces in the inner (left) and outer (right) sides of the GPETNC. $\Delta\psi$ indicates the drop between the potential far away from the graphene surface (where $\psi$ hardly changes) and the one on it. **d** $\Delta\psi$ inner and outer the graphene surface in 0.1 M KCl and CaCl$_2$ solutions at different voltages.

To test the hypothesis, we calculated the distributions of the ion concentration and electric potential in the GPETNC in chloride salt solutions at different applied voltages by numerically solving the Poisson-Nernst-Planck equations (Eqs. (4)–(6) in "Methods") using COMSOL Multiphysics 5.3a Software (COMSOL, Inc.), and the details of the model and boundary conditions are shown in Supplementary Fig. 7. It is difficult to obtain analytical solutions in nanochannels with complex structures, and thus numerical simulations with the finite element method were used here, which have also been successfully utilized in studying the ion transport in nanofluidics[9–11,22,27,30–32,39,40,61–68], including not only PETNCs[39,40,63–66] but also nanopores in 2D materials[9–11,22,27,67,68].

Our theoretical calculations indicated that indeed there were the voltage-dependent enrichment and depletion of ions inside the GPETNC, and the distribution of the potassium ion concentration $c_{K^+}$ in 0.1 M KCl solution is presented in Fig. 4a as an example. At negative voltages, $c_{K^+}$ near the tip of the GPETNC was larger than the bulk value (dashed line), indicating the enrichment of the potassium ions, while they were depleted near the base side. In contrast, at positive voltages, the ion depletion occurred near the tip side of the GPETNC, and the enrichment occurred near the base. The degrees of the ion enrichment/depletion both increased with the increase of the absolute value of the voltage, which is similar to the voltage dependence phenomenon in a pristine conical PETNC[39,40,63–66]. $c_{K^+}$ at all voltages increased sharply as the location approaches the graphene surface ($z = 12$ µm), which indicates that a great number of K$^+$ accumulated around there (Fig. 4b) owing to the attraction by the negative charges on the graphene subnanopores. The distribution of Cl$^−$ also relied on the voltage; it exhibited a similar behavior of enrichment/depletion to that of K$^+$, except for the pronounced depletion nearby the graphene surface (Supplementary Fig. 8a), and for other cation-chloride solutions,

similar ion distributions were observed (see Supplementary Fig. 8b–d for the results of 0.1 M CaCl$_2$ solution for example).

The extreme cation enrichment near the graphene surface could partially screen the negative charges there and reduce the effective surface charge density, which can be reflected by the variation of the electric potential $\psi$ around there[27,46]. As shown in Fig. 4c, the value of $\psi$ far away from the graphene is close to the applied voltage, and it decreases as the location approaches the graphene surface because of the negative charges. For the same electrolyte solution, the effective charge density of graphene surface $\sigma'_{gra}$ can be qualitatively reflected by the potential drop $\Delta\psi$ between the value of $\psi$ far away from the graphene surface and the value on the surface; a larger $\Delta\psi$ indicates a higher $|\sigma'_{gra}|$[27,46]. Figure 4d shows that $\Delta\psi$ in the outer side of the graphene surface barely changed at different voltages, because of the nearly unchanged degree of cation enrichment there (see Fig. 4b and Supplementary Fig. 8d). However, $\Delta\psi$ in the inner side decreased with the increment of applied voltage from −1 V to 1 V, especially for monovalent cations, which was caused by the higher degree of the cation accumulation near the graphene surface inner side at a larger voltage (Fig. 4b). The results indicated that the applied voltage affected the cation enrichment near the graphene subnanopores inside the GPETNC, and thus changed the effective surface charge density, making it rely on the voltage (as schematically illustrated in Fig. 5).

The ion selectivity of the GPETNC arose from its graphene subnanopores as discussed before. It has been reported that the value of charge density of graphene nanopores influences the ion selectivity[22,27,29,43,46,69]. Our previous molecular dynamics simulations demonstrated that a graphene subnanopore with negative charges had a higher ion selectivity than a pore with no charge[43], and things are similar for other subnanometer-sized channels (e.g., a single carbon nanotube with more negative charges has a higher ion selectivity[70]).

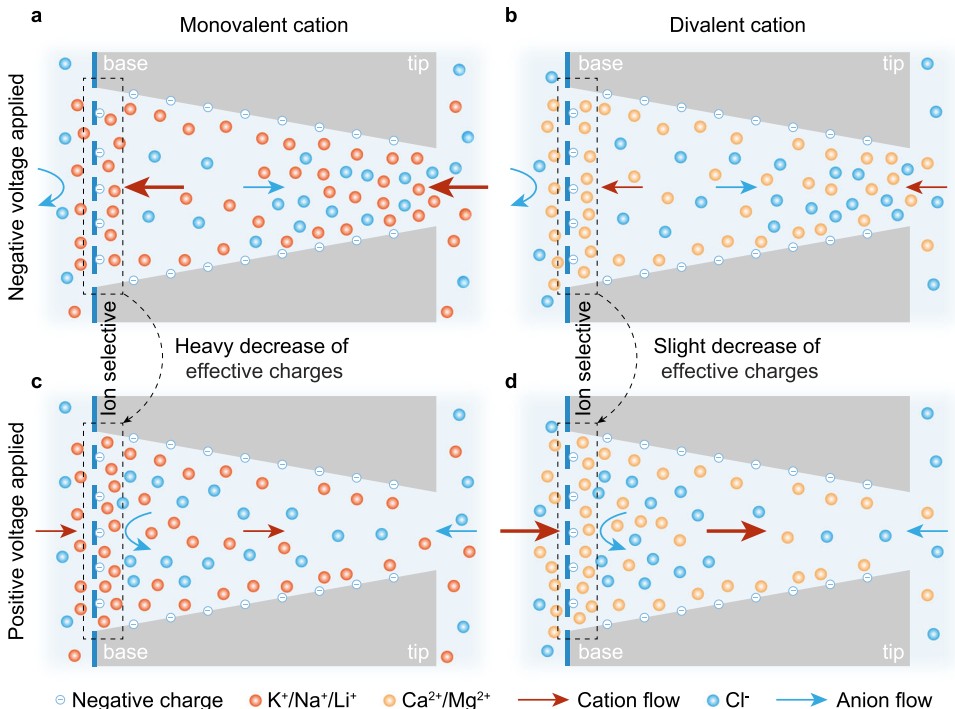

**Fig. 5 | Schematic illustration of the mechanism of ion sieving and rectification in graphene-based polyethylene terephthalate nanochannel (GPETNC).** The distribution and transport direction of anions and monovalent cations at negative voltages (**a**), divalent cations at negative voltages (**b**), monovalent cations at positive voltages (**c**), and divalent cations at positive voltages (**d**). Ion selective graphene subnanopores are on the base side of the conical polyethylene terephthalate nanochannel, where the voltage is applied. For monovalent cations (**a**, **c**), the graphene effective charges heavily decrease with the increase of applied voltage (curved arrow) due to the significant voltage-dependent cation enrichment near the graphene (dashed box). However, for divalent cations (**b**, **d**), the graphene effective charges slightly decrease with the increase of applied voltage (curved arrow), because the change of the cation enrichment degree is slight (dashed box).

Therefore, the inter-cation selectivity of the GPETNC relied on the applied voltage due to the voltage-dependent $\sigma'_{gra}$ demonstrated in the theoretical calculations. For monovalent cation-chloride solutions, the inner $\Delta\psi$ of the GPETNC at $-1$ V was much higher than that at 1 V (see Fig. 4d for KCl as an example), indicating a higher $|\sigma'_{gra}|$, and thus it made the Na$^+$ and Li$^+$ selectivities of the GPETNC at $-1$ V higher than those at 1 V, as observed in the experiments (Fig. 2c). Nevertheless, for divalent cation-chloride solutions (e.g., CaCl$_2$ in Fig. 4d), the change of $\Delta\psi$ between $-1$ V and 1 V was much smaller than that in the KCl solution, which indicates a weaker influence of the ion enrichment at different voltages on $\sigma'_{gra}$ as compared to the situations for monovalent ions (Fig. 5). Hence, the selectivities of Ca$^{2+}$ and Mg$^{2+}$ of the GPETNC slightly changed at different voltages. With the increase of applied voltage from $-1$ to 1 V, the selectivities of Na$^+$ and Li$^+$ decreased, and those of Ca$^{2+}$ and Mg$^{2+}$ kept in low values, eventually resulting in the high K$^+$/ions selectivity of the GPETNC at positive voltages as found in the experiments (Fig. 2c). It is worth noting that studies on the influences of graphene nanopore charges on the inter-cation selectivity were rare[43,71], and the mechanism of the charges influencing the ion transport was unclear (e.g., affecting the electrostatic repulsion or dehydration degree of ions)[3]. Such investigations are beyond the scope of this work, and thus future research can, for example, focus on the reasons why the experimentally measured ion selectivity of the GPETNC did not monotonously rely on the voltage, as departed from $\Delta\psi$ in theoretical calculations.

As for the opposite rectification of mono- and divalent ions of the GPETNC, it also arose from its heterogeneous structure. The GPETNC had a high K$^+$/Cl$^-$ selectivity of 46 as demonstrated before, which indicates that the transport of cations mainly contributes to the ion current. Theoretical calculations indicated the existence of the voltage-dependent cation enrichment/depletion inside the GPETNC (Fig. 4 and Supplementary Fig. 8), and the direction of the cation

concentration gradient is the same as that of the applied electric field, i.e., tip-to-base at negative voltages and base-to-tip at positive voltages, enhancing the ion current, as shown in Fig. 5. For monovalent cations, the absolute concentration gradient between the tip and base sides at negative voltages was higher than that at positive voltages (Fig. 4a), thus resulting in a larger diffusion current. In this case, the preferential transport direction of monovalent cations was tip-to-base, as the rectification ratios were characterized to be smaller than 1 (Fig. 2a). However, the situation for the divalent cations was different, i.e., the absolute concentration gradient between the tip and base sides at negative voltages was lower than that at positive voltages (Supplementary Fig. 8b). Thus, divalent cations had the preferential transport in the base-to-tip direction (rectification ratios larger than 1 as shown in Fig. 2a), which was opposite to that of monovalent cations (Fig. 5).

In conclusion, we fabricated a multifunctional heterogeneous GPETNC with facilely voltage-tunable ion sieving ability. The GPETNC exhibited the promise as an efficient mono/divalent ion selective device, an excellent biomimetic ion channel, and a cation-sensitive nanofluidic diode. Its ion selectivity is determined by the graphene subnanopores with negative charges, which are affected by the voltage-dependent enrichment/depletion of ions inside the GPETNC, and the opposite rectification of divalent ions compared to monovalent ones rely on the differences in the diffusion currents caused by the ion concentration gradients. The unique and fascinating performance of the GPETNC results from the interaction between the ion selective graphene subnanopores and the rectifying conical PETNC. Therefore, a device with similar heterogeneous structures (i.e., nanoporous 2D materials supported by rectifying nanochannels), suggested by our work, might also have a voltage-tunable ion selectivity even with a wider range and the cation-sensitive ion rectification with a more excellent rectification ratio. The creation of such composite can be facile and simple, because of the mature fabrication techniques of

NATMs[72–74] and ion rectifying nanochannels[30–32]. For example, a nanochannel with a much stronger rectifying ability, in which the degree of the ion enrichment/depletion is also heavier, could replace the original one of the GPETNC in this work, and thus this new nanochannel might have a much higher ion selectivity and rectification ratio. In addition, GO membranes with fixed interlayer spacing could be transferred onto a cylindrical PETNC with uneven surface charge distribution, which is another rectifying nanochannel[75]. In this case, one could tune the ion selectivity of this device by simply adjusting the applied voltage, with no need in overcoming the barriers in controlling the interlayer spacing of the pristine GO membranes. The heterogeneous nanochannel devices made by NATMs and traditional rectifying channels to achieve rapidly tunable ion selectivity for diverse applications can be a new and flourishing research avenue.

## Methods

### Growth of graphene

Monolayer graphene was grown on copper foil (25-μm-thick, annealed, uncoated, 99.8% (metal basis), Alfa Aesar (China) Chemical Co., Ltd) with a low-pressure chemical vapor deposition (CVD) method[41,42] using a graphene CVD system (Xiamen G-CVD Graphene Technology Co., Ltd). The Cu foil was successively cleaned via sonication in deionized water, acetone, and isopropyl alcohol (IPA) for 5 min to remove surface contaminants. After dried, the Cu foil was folded as a pocket with the open edges crimped to improve the quality of synthesized graphene in the interior surface of the pocket[76,77]. In the graphene growth, the Cu pocket was placed in a tube furnace and heated to 1030 °C for 30 min and maintained the temperature for 5 min in a mixture of 10 sccm $H_2$ and 10 sccm Ar. Then, 10 sccm $CH_4$ was additionally introduced for 20 min under a total pressure of ~0.30 Torr. The furnace was subsequently cooled down to ambient temperature in the mixture of 10 sccm $H_2$ and 10 sccm Ar for 40 min. Raman spectroscopy was utilized to evaluate the quality of the synthesized graphene, which was confirmed to be monolayer with negligible defects (Supplementary Fig. 2).

### Transfer of graphene

For characterizing graphene subnanopores, the CVD grown graphene was transferred onto a Quantifoil TEM grid (R 1.2/1.3, Au G200F1, Ted Pella, Inc.) using a similar direct transfer procedure reported in our previous work[43] before ion irradiation experiments. The TEM grid with the carbon side was placed onto the graphene grown on Cu foil, which was carefully pressed to make the graphene tightly contact the TEM grid. Two drops of IPA were gently dropped on the sample to further increase the adhesion between the graphene and TEM grid after the IPA evaporated. Subsequently, 1 M ammonium persulfate aqueous solutions were used to etch the Cu foil for approximately 50 min. The graphene on TEM grid was then rinsed in deionized water for 5 min by three times to remove residual etchants. After the cleanout, it was dried under an infrared baking lamp at temperature of ~60 °C for 10 min.

For measuring ion currents of the GPETNC, the CVD grown graphene was transferred onto a prepared PET membrane with a conically shaped nanochannel using a similar wet transfer procedure reported in our previous works[43,78] before ion irradiation experiments. Polymethyl methacrylate (PMMA, 950K A4, MicroChem Corp.) was spincoated onto the graphene grown on Cu foil with the first rotate speed of 500 rpm for 20 s and then 3000 rpm for 40 s. The sample was then placed on a heating plate (C-MAG HS 4, IKA Works Guangzhou Co., Ltd.) at temperature of 150 °C for 5 min to cure the PMMA layer. Similar to the transfer procedure of the graphene on TEM grid, 1 M ammonium persulfate aqueous solutions were used to etch the Cu foil for about 50 min, and the sample was rinsed in deionized water for 5 min by three times to remove residual etchants. Subsequently, the PMMA/graphene composite was transferred onto the prepared PET

membrane, and it was dried in ambient conditions for 30 min followed by desiccation under an infrared baking lamp at temperature of ~60 °C for 30 min. Trichloromethane was used to dissolve the PMMA of the sample for 18 h, which was then rinsed in IPA for 5 min to remove residuals. At last, the graphene supported on PET membrane was dried under an infrared baking lamp at temperature of ~60 °C for 30 min.

### Ion irradiation

To introduce subnanopores in graphene, the transferred graphene on TEM grid or PET membrane was perpendicularly irradiated by Au ions with energy of 500 keV and fluence of $1 \times 10^{13}$ cm$^{-2}$ (with current of ~30 nA) at room temperature and pressure of ~$2.2 \times 10^{-6}$ Torr using the $2 \times 1.7$ MV electrostatic accelerator of State Key Laboratory of Nuclear Physics and Technology, Peking University, China. To create a nanometric track (narrow damage trail) in PET membrane for subsequent track-etching procedure, a pristine PET membrane (12 μm thickness, Weifang Kaili Packing Materials Co., Ltd) was perpendicularly irradiated by a single $^{86}Kr^{26+}$ ion with energy of 2.15 GeV at room temperature using the sing-ion-hit system at the Lanzhou Interdisciplinary Heavy Ion Microbeam of Heavy Ion Research Facility in Lanzhou, Institute of Modern Physics, Chinese Academy of Sciences, China.

### Fabrication of PETNC

The PET membrane with a single conical nanochannel was fabricated by the asymmetric track-etching technique using a similar procedure reported in our previous works[39,40]. The prepared PET membrane with a nanometric track irradiated by a single swift heavy ion was chemically etched in a custom-designed system with one conductivity cell containing the etchant of 2 M NaOH and the other of the stop medium of 1 M HCOOH as well as 1 M KCl, which was heating in a water bath at temperature of 60 °C. A Keithley picoammeter (Keithley 6487, Keithley Instruments, Inc.) using gold electrodes was applied to monitor the ion current through the PET membrane in the etching procedure under the applied voltage of 1 V at the side of the etchant. When the ion current reached ~0.5 nA in ~10 min, which indicated the birth of a conical nanochannel, the etchant was replaced by the stop medium to terminate the etching process, and the sample was subsequently rinsed in deionized water to remove residuals.

### Characterization of graphene subnanopores

A dry-cleaning method with activated carbon powders[79–82] was utilized to reduce surface contaminations of graphene after ion irradiation experiments. Nanoporous graphene on TEM grid was embedded in activated carbon powders in a glass vial and heated on a heating plate (C-MAG HS 4, IKA Works Guangzhou Co., Ltd). The temperature of the sample was increased from room temperature to 200 °C with rate of 2.5 °C/min, and this temperature was held for 30 min. The sample was subsequently let to cool down to room temperature and carefully taken out of the glass vial, followed by being gently purged with a nitrogen stream to remove the residual activated carbon powders.

Before imaging, the dry-cleaned graphene on TEM grid was baked at temperature of 160 °C and pressure of $<10^{-5}$ Torr for at least 12 h to further reduce surface contaminations, allowed to cool down to about 60 °C under the vacuum, and then immediately loaded into the microscope column. Room temperature HAADF-STEM imaging was performed on the Nion U-HERMES200 aberration-corrected STEM located in the Electron Microscopy Laboratory of Peking University. The microscope was operated at an accelerating voltage of 60 kV to avoid electron radiation induced damage to the graphene while imaging[19,83–85]. The convergence semi-angel was 35 mrad with an annular recording range of 80-210 mrad. The electron probe diameters were focused to ~60 pm, and the beam current was measured as 28 pA for the imaging. The STEM images were recorded with $4096 \times 4096$ pixels, and each pixel collection time was 128 ms. The images were filtered using a Gaussian function implemented in the microscope

software, and the s-curves of the images were adjusted to increase the contrast between carbon atoms and subnanopores.

## Measurement of ion current

$I$–$V$ curves were measured by a Keithley picoammeter (Keithley 6487, Keithley Instruments, Inc.) using Ag/AgCl electrodes in a custom-designed system with two conductivity cells filled with chloride salt solutions at the same concentration or one cell filled with 1 M and the other with 1 mM KCl solutions for measuring cation/anion selectivity. Voltage was applied at the base side of the sample (Fig. 1b), and the scanning voltage was changed from −1 to +1 V with step of 0.2 V. Each $I$–$V$ curve was recorded three times to obtain the average and standard deviation of ion currents at corresponding voltages.

## Ion selectivity

Cation/anion (i.e., K$^+$/Cl$^-$ in this work) selectivity is identified using the Goldman–Hodgkin–Katz (GHK) model[50,51], which does not explicitly rely on the channel size and has been successfully utilized to characterize the ion selectivity of graphene nanopores[9,22,43]. Briefly, the K$^+$/Cl$^-$ selectivity $S_{GHK}$ is calculated according to the GHK voltage equation

$$V_{rev} = \frac{k_B T}{e} \ln\left(\frac{S_{GHK} c_{high} + c_{low}}{S_{GHK} c_{low} + c_{high}}\right) \quad (1)$$

where $k_B$ is the Boltzmann constant, $T$ is the temperature, $e$ is the elementary charge, $c_{high}$ and $c_{low}$ are the concentrations of KCl solutions at the two sides of the sample, and $V_{rev}$ is the reversal potential, which is the applied potential where the net current is zero. The $V_{rev}$ of the GPETNC was estimated to be 254 mV by cubic smoothing spline fitting of the measured $I$–$V$ curve at $c_{high}$ = 1 M and $c_{low}$ = 1 mM (Supplementary Fig. 4), and it was supposed to be calibrated in consideration of the redox reaction potentials on the Ag/AgCl electrodes in salt solutions, which was reported as 158.6 mV at such a concentration ratio of KCl solutions[86]. Therefore, the K$^+$/Cl$^-$ selectivity of the GPETNC was calculated to be about 46 based on Eq. (1).

Inter-cation selectivity is obtained by directly comparing the ion conductance in different chloride salt solutions at the same concentration, which is effective to describe the ability of ion channels to discriminate different cations under the circumstance that ion current due to Cl$^-$ can be neglected compared to that of cations[22,43]. In consideration of the different electric mobilities and charges of cations, a normalized conductance $g_i$ of cation i (compared to that of K$^+$) is defined[22,43] as

$$g_i = \frac{G_i}{Q_i \mu_i / \mu_{K^+}} \quad (2)$$

where $G_i$ is the measured channel conductance in the cation-chloride solution at different voltages, $Q_i$ is the electric charge of cation i, and $\mu_i$ is the electric mobility of cation i in bulk solutions. Hence, the inter-cation selectivity $S_i$ of cation i (relative to that of K$^+$) satisfies[22,43]

$$S_i = \frac{g_i}{g_{K^+}}. \quad (3)$$

## Numerical modeling

Ion transport in nanochannels is numerically calculated by solving the Poisson–Nernst–Planck (PNP) equations (Eqs. (4)–(6)) using COMSOL Multiphysics 5.3a Software (COMSOL, Inc.).

$$\nabla^2 \psi = -\frac{F}{\varepsilon} \sum_i z_i c_i \quad (4)$$

$$J_i = -D_i \left( \nabla c_i + \frac{z_i F}{RT} c_i \nabla \psi \right) \quad (5)$$

$$\nabla \cdot J_i = 0 \quad (6)$$

where $\psi$, $F$, $\varepsilon$, $R$, and $T$ are respectively the electric potential, Faraday constant, dielectric constant of the medium, ideal gas constant, temperature, and $z_i$, $c_i$, $J_i$, $D_i$ are respectively the electric charge, concentration, flux, and diffusion coefficient of ion i. The region to be solved and corresponding boundary conditions are shown in the schematic diagram of the model of GPETNC in Supplementary Fig. 7.

The theoretical model based on numerically solving the PNP equations described above is powerful to study the transport of charged species in ion channels. It is worth noting that there exist a few limitations for such a model, which include, for example, the neglect of steric hindrance caused by the finite volume of ions, dielectric effects such as partial dehydration, and weak van der Waals forces. However, those neglected effects can only quantitatively affect the calculated results, such as the values of ion concentration and electric potential, but they do not influence the correctness of the conclusions, because the electrostatic interaction and ion diffusion are nicely considered in the model.

## Data availability

The data that support the findings of this study are available from the corresponding author upon request.

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

## Acknowledgements

This work is supported by the National Natural Science Foundation of China (Grant Nos. 11775005 and 11975283), and the Science Challenge Project (No. TZ2018004). The authors thank the support from Heavy Ion Research Facility in Lanzhou. The authors acknowledge Electron Microscopy Laboratory of Peking University, China for the use of Cs-corrected Nion U-HERMES200 scanning transmission electron microscopy. The authors are grateful for the computing resources provided by the High Performance Computing Platform of the Center for Life Science of Peking University, and the Weiming No. 1 and Life Science No. 1 High Performance Computing Platform at Peking University.

## Author contributions

G.D. and J.X. conceived the research. S.S. performed the measurements of ion current and data analyses. Y.Z. performed the transfer of graphene and characterization using Raman spectroscopy. S.P. and Y.L. performed the numerical modeling and theoretical calculations. L.G. performed the ion irradiation of graphene and fabrication of PET nanochannels. E.F. performed the growth of graphene. H.Y. performed the ion irradiation of PET membranes. J.D. performed the cleaning and characterization of graphene subnanopores. All the authors discussed the results and prepared the manuscript.

## Competing interests

The authors declare no competing interests.
