## [Peer Review File · Nature Communications]

REVIEWER COMMENTS

Reviewer #1 (Remarks to the Author):

In this paper, Su et al. describe a novel voltage-tunable graphene-PET nano-micropore system which demonstrates a variety of interesting ion-selective behaviors, including cation selectivity, differences between rectification behavior of mono- and divalent cations, and very high selectivity for K⁺ ions. The authors have also carried out Multiphysics simulations which agree with their observed experimental results. Overall, this paper is a good addition to the body of literature of the active field of biomimetic ion-selective channels. I recommend publication with a minor revision noted below:

- In Figure 4, the Comsol figure shows extreme ion concentration gradients. In general, it is difficult to rationalize a concentration gradient at small length scales (ΔC of ~ 500 mM over 0.5 nm). Perhaps the authors should comment on the limitations of using such a model to depict this system, where the discreteness of ions, including the steric effects of hydration shells at this scale, can factor into the true concentrations, even if the apparent selectivity can justify the experimental observations.

Reviewer #2 (Remarks to the Author):

This manuscript reports a graphene-based polyethylene terephthalate nanochannel with voltage controllable ion sieving. At the same time, it exhibits ion current rectification where the monovalent and divalent cations prefer opposite flow directions. With questioning novelty, I feel that the argumentation of the results is not well constructed.

However, if this manuscript should be further considered, the following critical issues must be addressed.

1. I recommend an elaborate literature review for charge-guided ion transport in nanofluidic membranes in the introduction section. Authors missed out on some important literature showing excellent controllable ion transport in nanofluidic membranes. For example, "Controllable ion transport by surface-charged graphene oxide membrane," *Nat. Commun.* 10, 1253 (2019) and "Highly selective charge-guided ion transport through a hybrid membrane consisting of anionic graphene oxide and cationic hydroxide nanosheet superlattice units," *NPG Asia Mater.*, 8, e259 (2016).
2. The mechanism of highly selective transport of K⁺ ions as compared to that of other monovalent ions such as (Na⁺, Li⁺) at positive voltages needs an elaborate description in Figure 2c. I suggest studying the behavior of GPETNC in mixed electrolyte systems for better understanding as the biological system contains both K⁺ and Na⁺ ions.
3. A schematic explaining the mechanism of ionic current rectification and opposite flow of monovalent and divalent cations must be presented for a better understanding of readers.
4. The values for the ICR ratio reported in this manuscript are much lower than many reported in the literature, which questions the efficient rectification by GPETNC.
5. Plots of current as a function of time for fluctuating electric field should be recorded for monovalent and divalent cations to show the preferential direction of ion transport.

Reviewer #3 (Remarks to the Author):

This manuscript is about ion selectivity of heterogeneous graphene-based polyethylene terephthalate nanochannel (GPETNC) under an applied voltage. It demonstrates the combined effect of ion selective graphene subnanopores and the rectifying conical PETNC on the selectivity of mono- and divalent ions. The manuscript can be considered for publication after addressing the following.

- 1- The manuscript needs a major revision to improve the quality of writing and correct grammar

errors. In its current form, it is very difficult to understand some of the explanations and descriptions, which makes judging the research quality very difficult.

2- The ions selectivity of the membranes is demonstrated by changes in measured current under different applied potentials for different ions. A fundamental question here is if real selectivity from mixed ion electrolytes can be demonstrated. Can it be discussed that the selectivity based on the enrichment/depletion of specific ions will be the same in mixed ions solutions? Is there any way to experimentally demonstrate this?

3- The manuscript needs a more detailed explanation of the procedures and methods. In several instances, the paper has referred to other papers for methods which makes understanding of the process very difficult if the readers are not familiar with those references.

4- The discussion about the ion selectivity mechanism needs clarification. While the authors have performed simulations to explain the behavior of the membranes under applied voltage, the description of the theoretical work and its connection with the experimental results is not very clear. This might be because of the language problems mentioned above.functions?

Dear Reviewers,

Thanks for the insightful and kind suggestions concerning our manuscript. We have carefully considered the valuable comments and revised the manuscript with all the changes highlighted (for neatness, non-scientific changes for writing quality improvements and manuscript formatting requests are not marked). The point-to-point responses are as follows.

Point-to-Point Responses to the Reviewer's Comments

Reviewer #1:

Comments: In this paper, Su et al. describe a novel voltage-tunable graphene-PET nano-micropore system which demonstrates a variety of interesting ion-selective behaviors, including cation selectivity, differences between rectification behavior of mono- and divalent cations, and very high selectivity for K⁺ ions. The authors have also carried out Multiphysics simulations which agree with their observed experimental results. Overall, this paper is a good addition to the body of literature of the active field of biomimetic ion-selective channels. I recommend publication with a minor revision noted below.

Response: Thanks for the careful review of our manuscript. We appreciate the reviewer's valuable comments about the theme, content, and interest of our work. We have revised the manuscript according to the comments of the reviewer, which greatly helped us to improve the quality of the manuscript. The point-to-point responses are presented as follows.

Q1: In Figure 4, the Comsol figure shows extreme ion concentration gradients. In general, it is difficult to rationalize a concentration gradient at small length scales (deltaC of ~500 mM over 0.5 nm). Perhaps the authors should comment on the

limitations of using such a model to depict this system, where the discreteness of ions, including the steric effects of hydration shells at this scale, can factor into the true concentrations, even if the apparent selectivity can justify the experimental observations.

A1: Thanks for the insightful comment. Although theoretical calculations with the finite element method based on numerically solving the PNP equations are powerful to study the transport of charged species in ion channels, there indeed, as indicated by the reviewer, exist limitations for such a model, which include, for example, the neglect of steric hindrance caused by the finite volume of ions, dielectric effects such as partial dehydration, and weak van der Waals forces. In this work, those neglected effects can only quantitatively affect the calculated results, such as the values of ion concentration and electric potential, but they do not influence the correctness of the conclusions, because the electrostatic interaction and ion diffusion are nicely considered in the PNP model.

In order to clarify this issue, we added a discussion on the applicability of using the PNP model to depict the GPETNC in the revised manuscript, which is presented as follows.

Page 21, line 11 down from the top: *“The theoretical model based on numerically solving the PNP equations described above is powerful to study the transport of charged species in ion channels. It is worth noting that there exist a few limitations for such a model, which include, for example, the neglect of steric hindrance caused by the finite volume of ions, dielectric effects such as partial dehydration, and weak van der Waals forces. However, those neglected effects can only quantitatively affect the calculated results, such as the values of ion concentration and electric potential, but they do not influence the correctness of the conclusions, because the electrostatic interaction and ion diffusion are nicely considered in the model.”*

Reviewer #2:

Comments: This manuscript reports a graphene-based polyethylene terephthalate nanochannel with voltage controllable ion sieving. At the same time, it exhibits ion current rectification where the monovalent and divalent cations prefer opposite flow directions. With questioning novelty, I feel that the argumentation of the results is not well constructed. However, if this manuscript should be further considered, the following critical issues must be addressed.

Response: Thanks for reviewing our manuscript. We appreciate the reviewer's clear understanding of our work, and it is our honor to receive the reviewer's constructive and valuable comments on the argumentation of the results. We have revised the manuscript according to the comments from the reviewer, which greatly improved the quality of the manuscript. The detailed point-by-point responses are as follows.

Q1: I recommend an elaborate literature review for charge-guided ion transport in nanofluidic membranes in the introduction section. Authors missed out on some important literature showing excellent controllable ion transport in nanofluidic membranes. For example, "Controllable ion transport by surface-charged graphene oxide membrane," *Nat. Commun.* 10, 1253 (2019) and "Highly selective charge-guided ion transport through a hybrid membrane consisting of anionic graphene oxide and cationic hydroxide nanosheet superlattice units," *NPG Asia Mater.*, 8, e259 (2016).

A1: Thanks for the valuable comment. As suggested by the reviewer, we added several relevant references including the two mentioned above and others (e.g., *Science* 2017, **357**, 792-796; *Nat. Commun.* 2016, **7**, 11408) and corresponding discussions in the revised manuscript, which is presented in the following.

Page 3, line 14 down from the top: "Zhang *et al.*¹⁷ reported that the GO membranes with controllable surface charges introduced by polyelectrolyte modification exhibited various ion selectivity and charge-guided ion transport. Also, Sun *et al.*¹⁸ fabricated

hybrid GO-based membranes with different surface charges to achieve different charge-sensitive sieving of cations. However, in those works, a single functional membrane was actually fabricated in each case, and its ability to sieve ions was still untunable, which is the same for NATMs with diverse structures (e.g., interlayer spacing, pore diameter, surface charge, functional group at pore edge) created using different methods¹⁷⁻²¹.”

Q2: The mechanism of highly selective transport of K⁺ ions as compared to that of other monovalent ions such as (Na⁺, Li⁺) at positive voltages needs an elaborate description in Figure 2c. I suggest studying the behavior of GPETNC in mixed electrolyte systems for better understanding as the biological system contains both K⁺ and Na⁺ ions.

A2: Thanks for the insightful comment. We performed experiments to measure the ion current of GPETNC in mixed electrolyte solutions, and we further clarified the discussion about the mechanism of the ion sieving in the GPETNC, especially the high K⁺/ions selectivity as suggested by the reviewer, with the help of an added schematic illustration for a better understanding in the revised manuscript, which is presented in the following.

To study the performance of the GPETNC in mixed solutions, taking K⁺ and Na⁺ as an example, we fabricated a GPETNC in additional works, whose voltage-tunable ion selectivity (according to the experiments in single electrolyte solutions) is shown in Supplementary Fig. 6a (presented as follows). However, the following Supplementary Fig. 6b indicates that the *I-V* curve in the mixed solution containing both K⁺ and Na⁺ is complicated and suggests the elusive effect of the interaction between K⁺ and Na⁺ on the ion transport, which is difficult to be further investigated with existing methods. This is because all ions contribute to the total current and their portions are hard to be distinguished, which is the same for other nanochannels with selectivity characterized by the measured ion current based on the experiments of ion conductance or mobility. Previous research in single solutions focusing on the changes in ion conductance or

mobility has been reported to be effective to characterize the ion selectivity of nanochannels (e.g., *ACS Applied Materials & Interfaces* 2020, **12**, 24281-24288; *2D Materials* 2019, **7**, 015030; *Science* 2017, **358**, 511-513; *Nature Communications* 2016, **7**, 11408; *Nature Nanotechnology* 2015, **10**, 1053-1057). Therefore, the ion selectivity of the GPETNC in the manuscript is demonstrated based on the differences in measured ion current/conductance in the experiments in single electrolyte solutions.

It is worth noting that ion-selective nanochannels and membranes are designed and fabricated in order to function in mixed solutions, i.e., to separate a single (or more) ion from a mixture of ions, and such research is valuable for each emerging ion selective nanochannel. Therefore, we sincerely thank the reviewer for the valuable suggestion of the experiments of the GPETNC in mixed electrolyte solutions containing both K^+ and Na^+ ions, which helped us to clarify the mechanism of the ion selectivity of GPETNC. However, considering the difficulties in mixed electrolyte systems as mentioned above and that the experiments in single solutions focusing on the differences in ion conductance/mobility are effective to characterize the ion selectivity of nanochannels, which is suggested to be valid in mixed solutions (e.g., *ACS Applied Materials & Interfaces* 2020, **12**, 24281-24288; *2D Materials* 2019, **7**, 015030; *Science* 2017, **358**, 511-513; *Nature Communications* 2016, **7**, 11408; *Nature Nanotechnology* 2015, **10**, 1053-1057), detailed studies of the interaction between different ions in mixed solutions are beyond the scope of this work, and future research could focus on this, which is currently under the consideration in our next research.

As for the mechanism of the ion sieving in the GPETNC, the ion selectivity originates from the graphene subnanopores of the GPETNC, which is concluded by the heterogeneous structure of the GPETNC including the non-selective conical PETNC. It has been reported that the value of charge density of graphene nanopores influences the ion selectivity (e.g., *Nature Communications* 2016, **7**, 11408; *Nano Letters* 2019, **19**, 6400-6409; *ACS Applied Materials & Interfaces* 2020, **12**, 24281-24288; *Langmuir* 2020, **36**, 7400-7407). Our previous molecular dynamics simulations demonstrated that a graphene subnanopore with negative charges had a higher ion selectivity than a pore with no charge (*ACS Applied Materials & Interfaces* 2020, **12**, 24281-24288), and

things are similar for other subnanometer-sized channels, e.g., a single carbon nanotube with more negative charges has a higher ion selectivity (*The Journal of Chemical Physics* 2007, **126**, 084706). Therefore, the inter-cation selectivity of the GPETNC relied on the applied voltage due to the voltage-dependent effective charge density of graphene surface σ'_{gra} as demonstrated in the theoretical calculations. For monovalent cation-chloride solutions, the inner potential drop $\Delta\psi$ (the difference between the electric potential ψ far away from the graphene surface and the one on it) of the GPETNC at -1 V was much higher than that at 1 V (as shown in Fig. 4d in the manuscript for KCl as an example), indicating a higher $|\sigma'_{\text{gra}}|$, and thus it made the Na^+ and Li^+ selectivities of the GPETNC at -1 V higher than those at 1 V, as observed in the experiments. Nevertheless, for divalent cation-chloride solutions (e.g., CaCl_2 in Fig. 4d in the manuscript), the change of $\Delta\psi$ between -1 V and 1 V was much smaller than that in the KCl solution, which indicates a weaker influence of the ion enrichment at different voltages on σ'_{gra} as compared to the situations for monovalent ions (as illustrated in Fig. 5 in the manuscript, which is presented as follows). Hence, the selectivity of Ca^{2+} and Mg^{2+} of the GPETNC slightly changed at different voltages. According to the above discussion of the ion selectivity mechanism, with the increase of applied voltage from -1 to 1 V, the selectivities of Na^+ and Li^+ decreased, and those of Ca^{2+} and Mg^{2+} kept in low values, eventually resulting in the high K^+ /ions selectivity of the GPETNC at positive voltages as found in the experiments (Fig. 2c in the manuscript).

Thank the reviewer again for the comments. In order to clarify this issue and address the reviewer's concern, we revised the manuscript and the corresponding parts are presented as follows.

Page 7 in the revised Supplementary Information, Supplementary Figure 6: “**Studies on GPETNC in mixed electrolyte solutions.** **a** Ion selectivity of a GPETNC at different voltages. The presented results were from another GPETNC different from the one in the manuscript, which was fabricated to study the performance of GPETNC in mixed electrolyte solutions. **b** I-V curves of the GPETNC in (a) in 0.1 M KCl, 0.1 M NaCl, and 0.1 M KCl & 0.1 M NaCl solutions. Error bars were the standard deviations from the average of three independent records.”

Page 14, Figure 5: “**Schematic illustration of the mechanism of ion sieving and rectification in GPETNC.** The distribution and transport of mono-/di-valent cations and anions at negative/positive voltages are respectively illustrated in the four subfigures. Ion selective graphene subnanopores are on the base side of the conical PETNC, where the voltage is applied.”

Page 9, line 15 down from the top: “*The ion selectivity of the GPETNC was demonstrated based on the experiments in single electrolyte solutions. We noticed that ion-selective nanochannels and membranes are designed and fabricated in order to function in mixed solutions, i.e., to separate a single (or more) ion from a mixture of ions. Therefore, such research is valuable for each emerging ion selective nanochannel. However, for nanochannels with selectivities based on the experiments of ion conductance^{22,43,46} or mobility^{48,49}, which is characterized by the measured ion current, it is difficult to study their performance in mixed solutions, because all ions contribute to the total current and their portions are hard to be distinguished. Taking K^+ and Na^+ as an example, we fabricated a GPETNC with ion selectivity shown in Supplementary Fig. 6a, and the I-V curve in the mixed solution containing both K^+ and Na^+ was complicated and indicated the elusive effect of the interaction between K^+ and Na^+ on the ion transport (Supplementary Fig. 6b). Considering that the experiments in single solutions focusing on the differences in ion conductance/mobility are effective to characterize the ion selectivity of nanochannels and that the selectivity is suggested to be valid in mixed solutions^{22,43,46,48,49}, detailed studies of the interaction between different ions in mixed solutions are beyond the scope of this work, and future research could focus on this.*”

Page 13, line 8 up from the bottom: “*The results indicated that the applied voltage affected the cation enrichment near the graphene subnanopores inside the GPETNC, and thus changed the effective surface charge density, making it rely on the voltage (as schematically illustrated in Fig. 5).*”

Page 13, line 5 up from the bottom: “*The ion selectivity of the GPETNC arose from its graphene subnanopores as discussed before. It has been reported that the value of charge density of graphene nanopores influences the ion selectivity^{22,27,29,43,50,69}. Our previous molecular dynamics simulations demonstrated that a graphene subnanopore with negative charges had a higher ion selectivity than a pore with no charge⁴³, and things are similar for other subnanometer-sized channels (e.g., a single carbon nanotube with more negative charges has a higher ion selectivity⁷⁰). Therefore, the inter-cation selectivity of the GPETNC relied on the applied voltage due to the voltage-*

dependent σ'_{gra} demonstrated in the theoretical calculations. For monovalent cation-chloride solutions, the inner $\Delta\psi$ of the GPETNC at -1 V was much higher than that at 1 V (see Fig. 4d for KCl as an example), indicating a higher $|\sigma'_{gra}|$, and thus it made the Na^+ and Li^+ selectivities of the GPETNC at -1 V higher than those at 1 V, as observed in the experiments (Fig. 2c). Nevertheless, for divalent cation-chloride solutions (e.g., CaCl_2 in Fig. 4d), the change of $\Delta\psi$ between -1 V and 1 V was much smaller than that in the KCl solution, which indicates a weaker influence of the ion enrichment at different voltages on σ'_{gra} as compared to the situations for monovalent ions (Fig. 5). Hence, the selectivity of Ca^{2+} and Mg^{2+} of the GPETNC slightly changed at different voltages. With the increase of applied voltage from -1 to 1 V, the selectivities of Na^+ and Li^+ decreased, and those of Ca^{2+} and Mg^{2+} kept in low values, eventually resulting in the high K^+ /ions selectivity of the GPETNC at positive voltages as found in the experiments (Fig. 2c).”

Q3: A schematic explaining the mechanism of ionic current rectification and opposite flow of monovalent and divalent cations must be presented for a better understanding of readers.

A3: Thanks for the valuable comment. As suggested by the reviewer, we added a schematic illustration to explain the mechanism of the ion rectification in the GPETNC and also modified corresponding discussions in the revised manuscript, which are presented in the following.

Page 14, Figure 5: “**Schematic illustration of the mechanism of ion sieving and rectification in GPETNC.** The distribution and transport of mono-/di-valent cations and anions at negative/positive voltages are respectively illustrated in the four subfigures. Ion selective graphene subnanopores are on the base side of the conical PETNC, where the voltage is applied.”

Page 4, line 4 up from the bottom: “The opposite preferential transport directions of the mono- and di-valent cations arise from the differences in the diffusion currents, which are generated by the ion concentration gradients.”

Page 15, line 8 down from the top: “As for the opposite rectification of mono- and di-valent ions of the GPETNC, it also arose from its heterogeneous structure. The GPETNC had a high K^+/Cl^- selectivity of 46 as demonstrated before, which indicates that the transport of cations mainly contributes to the ion current. Theoretical calculations indicated the existence of the voltage-dependent cation enrichment/depletion inside the GPETNC (Fig. 4 and Supplementary Fig. 8), and the direction of the cation concentration gradient is the same as that of the applied electric field, i.e., tip-to-base at negative voltages and base-to-tip at positive voltages, enhancing the ion current, as shown in Fig. 5. For monovalent cations, the absolute concentration gradient between the tip and base sides at negative voltages was higher than that at positive voltages (Fig. 4a), thus resulting in a larger diffusion current. In

this case, the preferential transport direction of monovalent cations was tip-to-base, as the rectification ratios were characterized to be smaller than 1 (Fig. 2a). However, the situation for the divalent cations was different, i.e., the absolute concentration gradient between the tip and base sides at negative voltages was lower than that at positive voltages (Supplementary Fig. 8b). Thus, divalent cations had the preferential transport in the base-to-tip direction (rectification ratios larger than 1 as shown in Fig. 2a), which was opposite to that of monovalent cations (Fig. 5)."

Page 15, line 1 up from the bottom: *"...and the opposite rectification of divalent ions compared to monovalent ones rely on the differences in the diffusion currents caused by the ion concentration gradients."*

Q4: The values for the ICR ratio reported in this manuscript are much lower than many reported in the literature, which questions the efficient rectification by GPETNC.

A4: Thanks for the insightful comment. As indicated by the reviewer, the rectifying ability of the GPETNC in the manuscript is indeed weaker than some nanofluid diodes in previous literature. However, our work aims at reporting the potentials of the GPETNC as a multifunctional heterogeneous nanochannel with the voltage-tunable ion selectivity as well as the cation-sensitive rectification. Suggested by the GPETNC, it will be facile and simple to fabricate a device with similar heterogeneous structures (i.e., nanoporous 2D materials supported by rectifying nanochannels), which might have a much higher rectification ratio. Such a device is worth studying in future works.

In order to clarify this issue and address the reviewer's concern, we revised the manuscript as follows.

Page 16, line 9 down from the top: *"For example, a nanochannel with a much stronger rectifying ability, in which the degree of the ion enrichment/depletion is also heavier, could replace the original one of the GPETNC in this work, and thus this new nanochannel might have a much higher ion selectivity and rectification ratio."*

Q5: Plots of current as a function of time for fluctuating electric field should be recorded for monovalent and divalent cations to show the preferential direction of ion transport.

A5: Thanks for the suggestion. In rectifying nanochannels, the rectification ratio R obtained from the I - V curves (i.e., the ratio of the absolute ion currents at opposite voltages, which is defined as $|I_{+1V}|/|I_{-1V}|$ in the manuscript) is generally utilized to characterize the extent of the ion rectification. Furthermore, when compared to 1, R can also nicely reflect the preferential direction of ion transport, which has been reported in both our early works (e.g., *J. Phys. D: Appl. Phys.* 2007, **40**, 7077-7084; *ChemPhysChem* 2010, **11**, 859-864; *Phys. Chem. Chem. Phys.* 2011, **13**, 576-581) and other literature (e.g., *Chem. Soc. Rev.* 2010, **39**, 923-938; *Chem. Soc. Rev.* 2010, **39**, 1115-1132). Taking a common conical PET nanochannel with negative charges on the channel wall and $R = |I_{+1V}|/|I_{-1V}|$ as an example, when applying voltages to the base side of the PET nanochannel, the I - V curve shows $R < 1$ and indicates that the channel stays in the “on” state at negative voltages (where the absolute currents large than zero) and the “off” state at positive voltages (where the currents close to zero). At negative voltages, the applied electric field has the tip-to-base direction, and in such cases, the absolute currents being large than zero indicates that ion transport has the preferential direction the same as the electric field direction, i.e., from tip to base side. However, at positive voltages, the direction of the electric field reverses as base-to-tip, opposite to the preferential direction of ion transport, and therefore, the measured currents are smaller than those at negative voltages, i.e., $R < 1$. For other modified PET nanochannels with $R > 1$, it is also easy to conclude that the preferential direction of the ion transport is base-to-tip.

With regard to the GPETNC in the manuscript, the above discussions are also applicable. For monovalent ions (K^+ , Na^+ , and Li^+), their rectification ratios were smaller than 1, which indicated that the transport of these ions in the GPETNC had a preferential direction from the tip to the base side. However, for divalent ions (Ca^{2+} and Mg^{2+}), the preferential direction of transport reversed as base-to-tip, identified by the

rectification ratios larger than 1. The measured rectification ratios are sufficient to reflect the preferential direction of ion transport in the GPETNC. As for the ion current as a function of time for fluctuating electric field, its effectiveness to show the preferential direction of ion transport might be weakened by other effects, such as the capacitance effect in conical PET nanochannels. Therefore, in the present manuscript, the discussions of the preferential direction of ion transport in the GPETNC based on the rectification ratios are appropriate.

Thank the reviewer again for the kind suggestion. In order to clarify this issue and address the reviewer's concern, we revised the manuscript as follows.

Page 10, line 5 down from the top: *“The extent of the ion rectification is generally characterized by the rectification ratio (i.e., the ratio of the absolute ion currents at opposite voltages according to the I-V curves, which was defined as $|I_{+1V}|/|I_{-1V}|$ here), and its value also reflects the preferential direction of ion transport in rectifying nanochannels well^{29,30,56,57}.”*

Reviewer #3:

Comments: This manuscript is about ion selectivity of heterogeneous graphene-based polyethylene terephthalate nanochannel (GPETNC) under an applied voltage. It demonstrates the combined effect of ion selective graphene subnanopores and the rectifying conical PETNC on the selectivity of mono- and divalent ions. The manuscript can be considered for publication after addressing the following.

Response: Thanks for the careful review of our manuscript. We appreciate the reviewer's clear understanding and description of the manuscript. According to the professional comments from the reviewer, we have made corresponding modifications to the manuscript to improve its quality. The point-by-point responses are listed below.

Q1: The manuscript needs a major revision to improve the quality of writing and correct grammar errors. In its current form, it is very difficult to understand some of the explanations and descriptions, which makes judging the research quality very difficult.

A1: Thanks for the comment. We thoroughly revised the manuscript to address the writing issue. For neatness, these non-scientific changes are not highlighted.

Q2: The ions selectivity of the membranes is demonstrated by changes in measured current under different applied potentials for different ions. A fundamental question here is if real selectivity from mixed ion electrolytes can be demonstrated. Can it be discussed that the selectivity based on the enrichment/depletion of specific ions will be the same in mixed ions solutions? Is there any way to experimentally demonstrate this?

A2: Thanks for the insightful questions. Indeed, the ion selectivity of the GPETNC is demonstrated based on the differences in measured ion current/conductance in the experiments in single electrolyte solutions. As suggested by the reviewer, we performed experiments to study the performance of GPETNC in mixed electrolyte solutions

containing both K^+ and Na^+ as an example. We fabricated a GPETNC with voltage-tunable ion selectivity (according to the experiments in single electrolyte solutions) shown in Supplementary Fig. 6a (presented as follows). However, the following Supplementary Fig. 6b indicates that the I - V curve in the mixed solution containing both K^+ and Na^+ is complicated and suggests the elusive effect of the interaction between K^+ and Na^+ on the ion transport, which is difficult to be further investigated with existing methods. This is because all ions contribute to the total current and their portions are hard to be distinguished, which is the same for other nanochannels with selectivity characterized by the measured ion current based on the experiments of ion conductance or mobility. Previous research in single solutions focusing on the changes in ion conductance or mobility has been reported to be effective to characterize the ion selectivity of nanochannels (e.g., *ACS Applied Materials & Interfaces* 2020, **12**, 24281-24288; *2D Materials* 2019, **7**, 015030; *Science* 2017, **358**, 511-513; *Nature Communications* 2016, **7**, 11408; *Nature Nanotechnology* 2015, **10**, 1053-1057). Therefore, the ion selectivity of the GPETNC in the manuscript is demonstrated based on the differences in measured ion current/conductance in the experiments in single electrolyte solutions.

It is worth noting that ion-selective nanochannels and membranes are designed and fabricated in order to function in mixed solutions, i.e., to separate a single (or more) ion from a mixture of ions, and such research is valuable for each emerging ion selective nanochannel. Therefore, we sincerely thank the reviewer for the valuable questions about the performance of the GPETNC in mixed electrolyte systems, which helped us to clarify the mechanism of the ion selectivity of GPETNC. However, considering the difficulties in mixed electrolyte systems as mentioned above and that the experiments in single solutions focusing on the differences in ion conductance/mobility are effective to characterize the ion selectivity of nanochannels, which is suggested to be valid in mixed solutions (e.g., *ACS Applied Materials & Interfaces* 2020, **12**, 24281-24288; *2D Materials* 2019, **7**, 015030; *Science* 2017, **358**, 511-513; *Nature Communications* 2016, **7**, 11408; *Nature Nanotechnology* 2015, **10**, 1053-1057), detailed studies of the interaction between different ions in mixed solutions are beyond the scope of this work,

and future research could focus on this, which is currently under the consideration in our next research.

Thank the reviewer again for the questions. In order to clarify this issue and address the reviewer's concern, we revised the manuscript as follows.

Page 7 in the revised Supplementary Information, Supplementary Figure 6: “**Studies on GPETNC in mixed electrolyte solutions.** **a** Ion selectivity of a GPETNC at different voltages. The presented results were from another GPETNC different from the one in the manuscript, which was fabricated to study the performance of GPETNC in mixed electrolyte solutions. **b** I-V curves of the GPETNC in (a) in 0.1 M KCl, 0.1 M NaCl, and 0.1 M KCl & 0.1 M NaCl solutions. Error bars were the standard deviations from the average of three independent records.”

Page 9, line 15 down from the top: “The ion selectivity of the GPETNC was demonstrated based on the experiments in single electrolyte solutions. We noticed that ion-selective nanochannels and membranes are designed and fabricated in order to function in mixed solutions, i.e., to separate a single (or more) ion from a mixture of ions. Therefore, such research is valuable for each emerging ion selective nanochannel. However, for nanochannels with selectivities based on the experiments of ion conductance^{22,43,46} or mobility^{48,49}, which is characterized by the measured ion current, it is difficult to study their performance in mixed solutions, because all ions contribute to the total current and their portions are hard to be distinguished. Taking K⁺ and Na⁺ as an example, we fabricated a GPETNC with ion selectivity shown in Supplementary Fig. 6a, and the I-V curve in the mixed solution containing both K⁺ and Na⁺ was complicated and indicated the elusive effect of the interaction between K⁺ and Na⁺ on the ion transport (Supplementary Fig. 6b). Considering that the experiments in single

solutions focusing on the differences in ion conductance/mobility are effective to characterize the ion selectivity of nanochannels and that the selectivity is suggested to be valid in mixed solutions^{22,43,46,48,49}, detailed studies of the interaction between different ions in mixed solutions are beyond the scope of this work, and future research could focus on this.”

Q3: The manuscript needs a more detailed explanation of the procedures and methods. In several instances, the paper has referred to other papers for methods which makes understanding of the process very difficult if the readers are not familiar with those references.

A3: Thanks for the kind suggestion. We revised the manuscript with a clearer description of the fabrication procedures of the GPETNC as follows, and more details (e.g., instrument models and parameter settings) are presented in the Methods section of the manuscript.

Page 6, line 1 down from the top: *“An asymmetric track-etching technique^{37,38} was used to fabricate the conical nanochannel in a 12- μ m-thick PET membrane. Briefly, a pristine PET membrane was first irradiated by a single swift heavy ion, which could create a nanometer-sized damage track in the membrane, and then asymmetrically etched with one side of the membrane contacting the etchant and the other contacting the stop medium. The fabricated conical PETNC had the base entrance diameter of \sim 840 nm estimated from the etching time and the tip diameter of \sim 140 nm obtained by fitting the size of a theoretical model to experimental data (Supplementary Fig. 1). Monolayer graphene grown by a chemical vapor deposition method^{39,40} with negligible defects (Supplementary Fig. 2) was transferred on the prepared PET membrane (making graphene on the base side of the PETNC) using a wet transfer procedure with polymethyl methacrylate as the mediator^{9,41}. Ultimately, irradiation of energetic ions^{9,41} was utilized to introduce subnanopores in the transferred graphene suspended on the PETNC. More details about the fabrication procedures of the GPETNC are presented in Methods.”*

Q4: The discussion about the ion selectivity mechanism needs clarification. While the authors have performed simulations to explain the behavior of the membranes under applied voltage, the description of the theoretical work and its connection with the experimental results is not very clear. This might be because of the language problems mentioned above.

A4: Thanks for the valuable comment. In response to the writing issues, we thoroughly revised the manuscript to address them, especially for the mechanism discussion, and those non-scientific changes are not highlighted for neatness. Moreover, as suggested by the reviewer, we further clarified the discussion about the mechanism of the ion sieving in the GPETNC and added a schematic illustration for a better understanding, which are as follows. As for the ion selectivity mechanism and the connection between the theoretical work and the experiments, briefly, the ion selectivity of the GPETNC originates from its graphene subnanopores (concluded by the heterogeneous structure of the GPETNC), whose effective charge density is affected by the nearby cation enrichment and depends on the applied voltage (demonstrated by the theoretical calculations). Graphene subnanopores with different charges have different ion selectivity (reported by our previous MD simulations and other references), and therefore, the GPETNC has voltage-dependent ion selectivity (demonstrated by the experiments). More detailed discussions are presented in the following.

Page 14, Figure 5: “**Schematic illustration of the mechanism of ion sieving and rectification in GPETNC.** The distribution and transport of mono-/di-valent cations and anions at negative/positive voltages are respectively illustrated in the four subfigures. Ion selective graphene subnanopores are on the base side of the conical PETNC, where the voltage is applied.”

Page 13, line 8 up from the bottom: “The results indicated that the applied voltage affected the cation enrichment near the graphene subnanopores inside the GPETNC, and thus changed the effective surface charge density, making it rely on the voltage (as schematically illustrated in Fig. 5).”

Page 13, line 5 up from the bottom: “The ion selectivity of the GPETNC arose from its graphene subnanopores as discussed before. It has been reported that the value of charge density of graphene nanopores influences the ion selectivity^{22,27,29,43,50,69}. Our previous molecular dynamics simulations demonstrated that a graphene subnanopore with negative charges had a higher ion selectivity than a pore with no charge⁴³, and things are similar for other subnanometer-sized channels (e.g., a single carbon nanotube with more negative charges has a higher ion selectivity⁷⁰). Therefore, the inter-cation selectivity of the GPETNC relied on the applied voltage due to the voltage-dependent σ'_{gra} demonstrated in the theoretical calculations. For monovalent cation-

chloride solutions, the inner $\Delta\psi$ of the GPETNC at -1 V was much higher than that at 1 V (see Fig. 4d for KCl as an example), indicating a higher $|\sigma'_{gra}|$, and thus it made the Na^+ and Li^+ selectivities of the GPETNC at -1 V higher than those at 1 V , as observed in the experiments (Fig. 2c). Nevertheless, for divalent cation-chloride solutions (e.g., CaCl_2 in Fig. 4d), the change of $\Delta\psi$ between -1 V and 1 V was much smaller than that in the KCl solution, which indicates a weaker influence of the ion enrichment at different voltages on σ'_{gra} as compared to the situations for monovalent ions (Fig. 5). Hence, the selectivity of Ca^{2+} and Mg^{2+} of the GPETNC slightly changed at different voltages. With the increase of applied voltage from -1 to 1 V , the selectivities of Na^+ and Li^+ decreased, and those of Ca^{2+} and Mg^{2+} kept in low values, eventually resulting in the high K^+ /ions selectivity of the GPETNC at positive voltages as found in the experiments (Fig. 2c).”

REVIEWERS' COMMENTS:

Reviewer #2 (Remarks to the Author):

1. The reviewer recommended an elaborate literature review for charge-guided ion transport in nanofluidic membranes in the introduction section and it has been addressed by the authors.
2. The reviewer suggested elaborate explanation of the mechanism of ionic current rectification and opposite flow of monovalent and divalent cations. The authors addressed this with an elaborate explanation and schematic representation.
3. I questioned rectifying ability of GPETNC. Authors agreed to the weaker rectification of GPETNC as compared to other nanofluid diodes. However, they pointed out that GPETNC is a multifunctional heterogenous nanochannel with "voltage-tunable ion selectivity" as well as "cation-sensitive rectification". Hence, this opens up research opportunity for development of heterogenous nanochannels with higher ion selectivity and rectification ratio.
4. I suggested experimental demonstration of selective ion transport in the mixed ion solutions. Reviewer 3 has also similar comment on the system. Though, authors tried to experimentally demonstrate the performance of GPETNC in mixed electrolyte solutions, they fail to demonstrate it due partly to interaction between different ions in mixed solutions. However, the ion selectivity is demonstrated based on single electrolyte solutions.

I feel that the authors have elaborately explained the GPETNC system for voltage controllable ion sieving in the revised version for a better understanding of the readers. Based on the aforementioned considerations, I recommend this revised manuscript for publication.